# Vision Foundation Models Bridge the Geometric Knowledge Across Domains for Long-Tailed Recognition

## Abstract

Deep learning struggles to fully unleash its potential in scenarios with limited sample sizes, primarily because models fail to capture information beyond the observed domain when the number of samples from rare classes is limited. Therefore, restoring the true distribution of rare classes becomes a significant challenge. In this study, we discovered that vision foundation models can associate inter-class similarity with the similarity of geometric shapes of class distributions in cross-domain scenarios. Specifically, we observed that when two cross-domain classes are highly similar, their embedding distributions also exhibit similar geometric shapes and sizes. These phenomena only manifest when using foundation models to represent images. Our findings provide a foundation for leveraging geometric knowledge of existing data distributions to assist rare classes. Further, we propose the Geometrically Guided Uncertainty Representation (GUR) Layer tailored for long-tailed recognition tasks, aiming to calibrate and augment the embedding distribution of tail classes, thereby learning an unbiased MLP classifier. Across multiple long-tailed benchmark datasets, GUR significantly enhances the performance of vision foundation models and achieves state-of-the-art results on certain datasets. The success of GUR serves as a typical example of integrating and colliding foundation models with prior knowledge.

## 1 Introduction

Learning from long-tailed data is one of the most common challenges in practical computer vision, as models often perform poorly on classes with sparse samples and exhibit significant bias. Intuitively, the lack of samples seems to be the source of model bias, but recent studies have suggested that models do not always perform poorly on tail classes Ma et al. (2023a); Sinha et al. (2022); Ma et al. (2023b). Chu et al. (2020) and Ma et al. (2024b) suggest that this may be related to whether the sparse samples in the tail classes cover their true distribution well. As shown in Figure 1A, when a small number of samples from tail classes can cover the true distribution well (red distribution), existing methods such as class re-balancing Lin et al. (2017); Cui et al. (2019); Wang et al. (2020b); Tan et al. (2021); Sinha et al. (2022); Ma et al. (2023a); Ren et al. (2020); Ye et al. (2020); Zhang & Pfister (2021); Alshammari et al. (2022); Han et al. (2005); Wang et al. (2019); Kang et al. (2019), data augmentation Zang et al. (2021); Zhong et al. (2021); Li et al. (2021); Zhang et al. (2017); Yun et al. (2019), and module improvement Zhou et al. (2020); Wang et al. (2020c); Huang et al. (2016); Dong et al. (2017); Kang et al. (2020); Cui et al. (2021); Ouyang et al. (2016); Cai et al. (2021); Wang et al. (2020a) only need to overcome gradient imbalance Tan et al. (2021) to effectively improve the model's performance on tail classes. However, when the sparse samples from tail classes are not representative enough (blue distribution), the model fails to fully learn the information of the true distribution. Therefore, additional knowledge is needed to help tail classes recover their true distribution as much as possible.

Research on knowledge transfer for long-tailed recognition Chu et al. (2020); Yin et al. (2019); Liu et al. (2020); Kim et al. (2020); Liu et al. (2021); Park et al. (2022); Cui et al. (2018); Yang & Xu (2020); Hu et al. (2020) in the past has focused on transferring knowledge from head classes to tail classes. For example, combining background information from head class samples with tail class samples to generate new samples in both image space Park et al. (2022) and embedding space

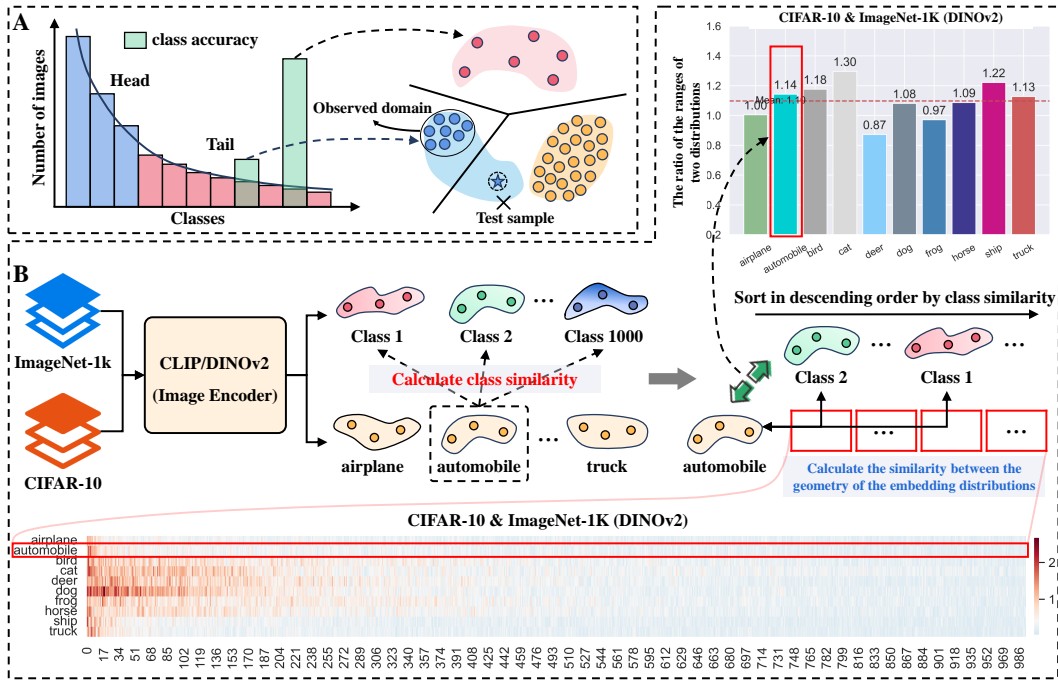

Figure 1: A. The model may not always perform poorly on tail classes; this depends on whether the observed distribution adequately covers the true distribution. B. Utilizing the foundation model to represent all classes across different datasets, we calculate the similarity between classes from different datasets and the similarity of geometric shapes and sizes between embedding distributions.

Chu et al. (2020), and using the variance of head classes to expand the distribution of tail classes Ma et al. (2024a). Their fundamental assumption is that head classes and tail classes have the same background information or distribution statistics, but lack direct evidence to support it. Recently, FUR Ma et al. (2024b) defined the geometric shape of distributions and found that when training datasets with smaller models such as ResNet, if two classes in the feature space are more similar (these classes come from the same dataset), their geometric shapes are also more similar. Therefore, the geometric shape of head classes can be used to guide the recovery of the true distribution of tail classes, but its application scenarios have obvious limitations:

(1) The number of head classes in long-tail datasets accounts for only a small fraction of the total number of classes, resulting in only a few choices when matching similar classes for tail classes, and there may not be classes in head classes that are most similar to tail classes.

(2) When the number of samples for all classes in the dataset is small, the geometric shape of the distribution is inaccurate and cannot be used as available knowledge.

Therefore, we further consider if it is possible to match and transfer geometric knowledge of similar classes from outside the long-tail dataset to help tail classes recover the true distribution, there will be more choices. Unfortunately, we found that small models cannot associate the geometric shapes of embedding distributions across datasets (Section 2.2.1).

In this study, we consider the strong representational capabilities of Vision Foundation Models. Therefore, we attempt to use CLIP Radford et al. (2021) and DINOv2 Oquab et al. (2023) to associate the geometric shapes of embedding distributions across datasets (as illustrated in Figure 1B). Surprisingly, we discover the following phenomena between the embedding distributions of different datasets represented by CLIP and DINOv2 (Section 2.2.2):

(1) The more similar the two categories are, the more similar the geometric shapes of their corresponding embedding distributions tend to be.

(2) The sizes of embedding distributions of two categories with high similarity are also close.

(3) For the same category, the matched category with high similarity is the same for two cases of sparse and sufficient samples.

These phenomena enable us to transfer geometric knowledge of embedding distributions across datasets to help categories with limited samples recover their true distribution as much as possible. An extreme example is that even when each class contains only one sample, distribution expansion can still be achieved through transferring geometric knowledge (see Section 4.5). Our findings could provide potential tools for other domains, such as few-shot learning and federated learning Shi et al. (2024); Chen et al. (2024).

Specifically, in Section 3.1, we propose leveraging geometric knowledge to generate new samples for tail classes, thereby restoring and calibrating the embedding distributions of the tail classes. Then, we utilize the calibrated embedding distributions to train an unbiased MLP as a long-tail classifier. However, the drawback is that generating new samples interrupts the training process, and the MLP can only be trained after the sample generation is complete. To achieve end-to-end training, in Section 3.2, we introduce the Geometric Uncertainty Representation (**GUR**) layer guided by geometric knowledge. The embeddings belonging to tail classes are transformed into uncertain embeddings through the GUR layer, which better represents the true distribution of tail classes. **It is worth noting that: (1)** The GUR layer we propose does not directly generate samples but serves as a plug-and-play module, which is convenient and easy to use. **(2)** Our method does not involve fine-tuning the Vision Foundation Models; the few learnable parameters come solely from the MLP network (See comparisons in Section 4.5). Experimental results on multiple long-tailed benchmark datasets demonstrate that our method significantly improves the performance of Vision Foundation Models in long-tailed scenarios (See Section 4.3 and 4.4).

## 2 VISION FOUNDATION MODEL AS BRIDGES FOR TRANSFERRING GEOMETRIC KNOWLEDGE

Transferring the geometric shape of distributions to help rare classes recover their true distribution requires consideration of three core issues: **(1)** Whether the geometric shape being transferred is similar to the true distribution of the tail class. **(2)** Whether the size of the transferred distribution is close to the size of the true distribution of the tail class, as size will affect whether the reconstructed tail class distribution can cover its true distribution well. **(3)** For a class, whether the matched classes with high similarity are the same in the two cases of sparse and sufficient samples, respectively. For the first two issues, we use the complete versions of CIFAR-10-LT and CIFAR-100-LT to obtain the true distributions of tail classes (Section 2.2). For the third issue, we will explore using both CIFAR-10-LT and CIFAR-100-LT, as well as their complete versions (Section 2.3).

### 2.1 GEOMETRIC SHAPE, SIZE, AND SIMILARITY OF EMBEDDING DISTRIBUTIONS

The geometric shape of data distributions can be represented by the eigenvectors of the covariance matrix Ma et al. (2024b). Specifically, in a $P$-dimensional space, given a class of data $X = [x_1, \ldots, x_n] \in \mathbb{R}^{P \times n}$, the covariance matrix of the distribution can be estimated as

$$\Sigma_X = \mathbb{E}\left[\frac{1}{n}\sum_{i=1}^{n} x_i x_i^T\right] = \frac{1}{n}XX^T \in \mathbb{R}^{P \times P}. \tag{1}$$

Performing eigenvalue decomposition on $\Sigma_X$ yields $P$ eigenvalues $\lambda_1 \geq \cdots \geq \lambda_P$ and their corresponding $P$-dimensional eigenvectors $\xi_1, \ldots, \xi_P$. All eigenvectors are mutually orthogonal, anchoring the skeleton of the data distribution. Thus, the geometric shape of data $X$ is defined as $GD_X(\xi_1, \ldots, \xi_P)$. Considering each eigenvalue represents the range of the distribution along the direction of the corresponding eigenvector, the **size of the distribution** can be measured by the sum of the eigenvalues, defined as

$$S(X) = \sum_{i=1}^{P} \lambda_i. \tag{2}$$

Given the geometric shapes of two distributions, $GD_{X_1}(\xi_{X_1}^1, \ldots, \xi_{X_1}^P)$ and $GD_{X_2}(\xi_{X_2}^1, \ldots, \xi_{X_2}^P)$, their similarity is defined as

$$S(GD_{X_1}, GD_{X_2}) = \sum_{i=1}^{P} \left|\langle \xi_{X_1}^i, \xi_{X_2}^i \rangle\right|. \tag{3}$$

The range of $S(GD_{X_1}, GD_{X_2})$ is $[0, P]$, where a larger value indicates greater geometric similarity. In this study, we follow the same setup as Ma et al. (2024b) by using the eigenvectors corresponding to the top five eigenvalues to compute the similarity of geometric shapes.

## 2.2 TRANSFERABILITY OF GEOMETRIC SHAPES OF EMBEDDING DISTRIBUTIONS

### 2.2.1 SMALL MODELS UNABLE TO ASSOCIATE GEOMETRIC SHAPES ACROSS DATASETS

We train a standard ResNet-34 He et al. (2016) on CIFAR-10 Krizhevsky et al. (2009) and then extract $p$-dimensional image embeddings from the last hidden layer of ResNet-34 for each class. Suppose the embedding set for class $i$ is $Z_i = [Z_i^1, \ldots, Z_i^n] \in \mathbb{R}^{p \times n}$, and the distribution center of $Z_i$ is $O_i = \frac{1}{n}\sum_{k=1}^{n} Z_i^k \in \mathbb{R}^{p \times 1}$. Given the embedding set $Z_j$ for class $j$, the similarity between class $i$ and class $j$ is calculated as $\frac{O_i^T O_i}{\|O_i\| \cdot \|O_j\|}$, where a larger value indicates greater similarity. First, compute the similarity between each class and other classes, and then sort the classes in descending order based on their similarities. Then, compute the similarity of the geometric shapes of embedding distributions between each class and other classes in sequence according to the class similarity. The experimental results are plotted in Figure 2A. It can be observed that as the class similarity increases, the similarity of geometric shapes also tends to increase. We also verify this phenomenon in the same experiments conducted on CIFAR-100 dataset. However, we found that when matching similar classes across datasets and calculating the similarity of geometric shapes of embedding distributions, there is no phenomenon where similar classes have similar geometric shapes.

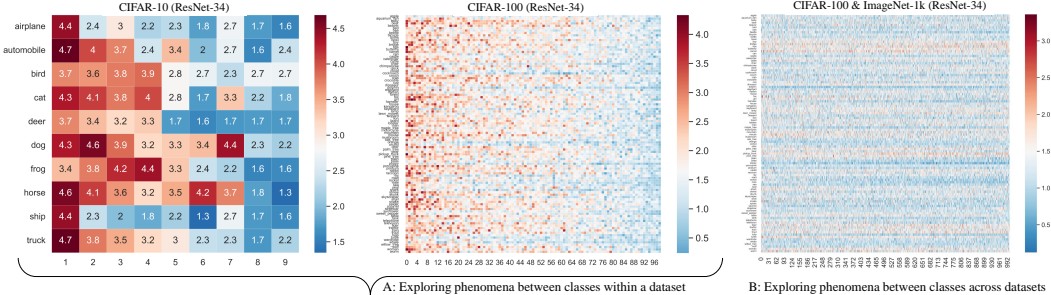

Figure 2: The horizontal coordinate is the index of classes, indicating from left to right the classes that are most similar to the class in the vertical coordinate to the least similar, respectively. Each element represents the similarity between the geometric shapes of the embedding distributions.

We train ResNet-34 on CIFAR-100 and ImageNet-1k Russakovsky et al. (2015) respectively, then extract image embeddings from the last hidden layer of ResNet-34. Next, we calculate the similarity between each class in CIFAR-100 and all classes in ImageNet and sort the classes in CIFAR-100 in descending order based on their similarities. According to the sorted order, we compute the similarity of the geometric shapes of embedding distributions between each class in CIFAR-100 and each class in ImageNet. However, the experimental results in Figure 2B do not exhibit the same pattern as Figure 2A.

### 2.2.2 FOUNDATION MODELS: BRIDGING GEOMETRIC KNOWLEDGE ACROSS DATASETS

The preceding experiments have demonstrated that small models cannot associate geometric knowledge across datasets. Given the powerful representational capabilities of Vision Foundation Models, we aim to leverage them for cross-dataset geometric knowledge transfer. Below, we primarily focus on two aspects: **(1)** Within a single dataset, do Vision Foundation Models exhibit the same phenomena as small models? **(2)** Can Vision Foundation Models also demonstrate a phenomenon where, for two classes belonging to different datasets, the more similar they are, the more similar their embedding distribution geometric shapes tend to be?

We selected CLIP and DINOv2 (ViT-B/16) as the two Vision Foundation Models to investigate the transferability of geometric shapes. Firstly, we extracted image embeddings from CIFAR-100 and Caltech-101 Fei-Fei et al. (2004) using CLIP and DINOv2 separately. Subsequently, we conducted experiments similar to those in Figure 2, wherein we matched similar classes within each dataset and computed the similarity of geometric shapes between the embedding distributions of corresponding classes. The experimental results are depicted in Figure 3, showing that Vision Foundation Models also exhibit the phenomenon of geometric shapes of embedding distributions becoming more similar as classes become more similar within datasets. It is noteworthy that compared to CLIP, DINOv2 shows a more pronounced phenomenon. This may be attributed to DINOv2 being trained through

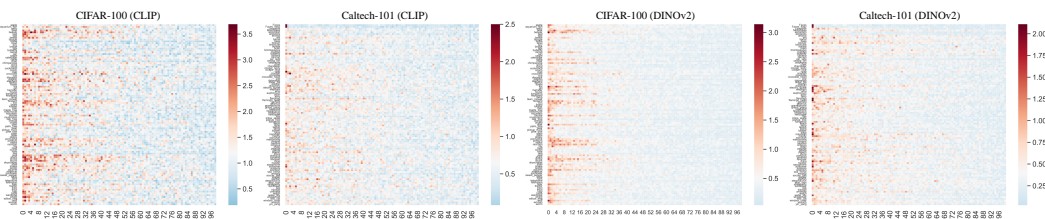

Figure 3: Calculate the similarity between classes within the dataset and the similarity of geometric shapes between corresponding embedding distributions.

multi-level masked self-supervised learning from pure visual data. It pays attention to finer details of visual information, enabling a more refined representation of the geometric shapes of embedding distributions for each class, thus making the phenomenon more evident.

Further, we selected ImageNet-1k as an external knowledge base and calculated the similarity between each class in CIFAR-100 and Caltech-101 with all classes in ImageNet-1k, as well as the similarity of geometric shapes between the embedding distributions corresponding to classes. Following the same procedure as Figure 3, we plotted the experimental results in Figure 4. To our surprise, we found that as the similarity between classes increased, the similarity of geometric shapes between the embedding distributions of corresponding classes also tended to increase. This suggests that Vision Foundation Models can serve as a bridge for associating geometric knowledge across datasets. Particularly, the phenomenon observed with DINOv2 was more pronounced, leading us to speculate that the geometric shapes represented by DINOv2 could more effectively and accurately guide the recovery of the true distribution of tail classes.

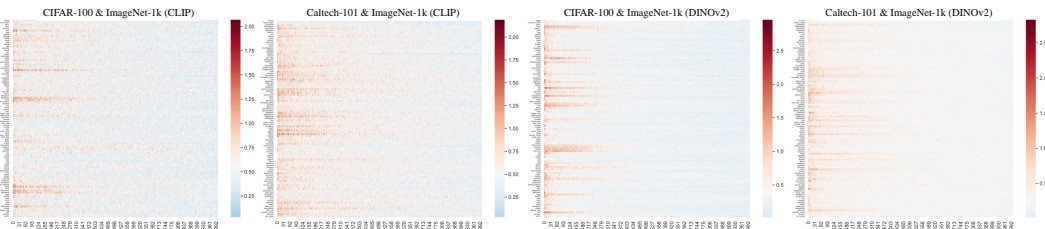

Figure 4: Compute the similarity between classes belonging to different datasets and the similarity of geometric shapes between corresponding embedding distributions.

When assisting tail classes, it's also important to consider the size of the distribution being transferred. We matched each class from CIFAR-100 and Caltech-101 with the most similar class from ImageNet-1k, and then computed the ratio of sizes between the embedding distributions corresponding to these two classes. The experimental results, as shown in Figure 5, reveal that the distributions represented by CLIP and DINOv2 exhibit similar sizes for embedding distributions of similar classes. The above results will support our naturally proposing a recovery method for tail class distributions.

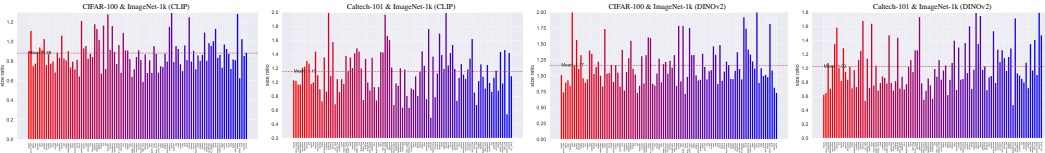

Figure 5: For two classes from different datasets with high similarity, the ratio of their embedding distributions (represented by the foundation model) sizes approaches 1.

## 2.3 MATCHING SIMILAR CLASSES FOR TAIL CLASS DISTRIBUTION RESTORATION

The study above was conducted under the assumption of sufficient samples, focusing on the geometric similarity between real distributions. Assuming we want to restore the true distribution of tail classes in CIFAR-10-LT, an ideal approach would be to match the tail classes with the most

similar classes in ImageNet-1k, and then transfer the geometric shape and size of the embedding distribution corresponding to those classes to the tail classes. However, a crucial prerequisite for this approach's feasibility is that the highly similar classes matched for tail classes in ImageNet-1k are consistent with those matched when samples are sufficient.

We still use CLIP and DINOv2 to extract the embedding distributions of the classes. We match the highest similarity classes $C_1^1, \ldots, C_{40}^1$ in ImageNet-1k for the 40 tail classes $C_1, \ldots, C_{40}$ with the least samples in CIFAR-100-LT, respectively. At the same time, we find the complete versions $T_1, \ldots, T_{40}$ of the 40 tail classes in CIFAR-100, and then match them with the first, second, and third most similar classes $T_i^1, T_i^2, T_i^3, i = 1, \ldots, 40$ in ImageNet-1k. We check whether $C_i^1$ matches $T_i^1, i = 1, \ldots, 40$, and calculate the proportion of matches out of 40, with the experimental results plotted in Figure 6. The ideal scenario is for $C_1^1, \ldots, C_{40}^1$, and $T_1^1, \ldots, T_{40}^1$ to be completely consistent, with a proportion of $100\%$. However, this is quite a strict requirement. For a class, the geometric shape of the second and third most similar classes also has high similarity with its geometric shape, so we relax the requirement. We calculate the proportion of $C_i^1$ contained in $T_i^1$ and $T_i^2, i = 1, \ldots, 40$, and whether $C_i^1$ is contained in $T_i^1, T_i^2, T_i^3, i = 1, \ldots, 40$. The experimental results in Figure 6 show that the most similar classes matched for tail classes from external datasets are also highly similar classes corresponding to sufficient samples of tail classes. This ensures the feasibility and reliability of finding and transferring geometric knowledge based on class similarity.

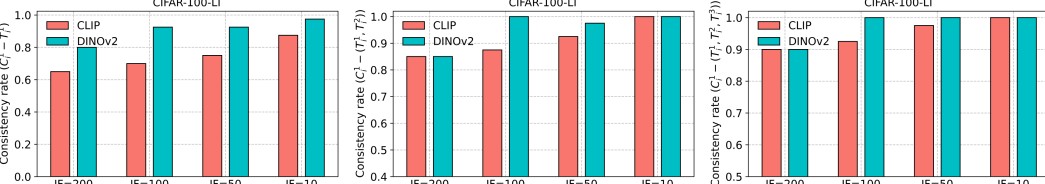

Figure 6: Match the most similar classes in ImageNet-1k for the 40 tail classes of CIFAR-100-LT as well as their full versions, and calculate the proportion of tail classes that agree with the most similar class matched to their full versions. A schematic of the above process is in Appendix B.

## 3 TAIL CLASS DISTRIBUTION CALIBRATION WITH FOUNDATION MODEL

In this section, we first introduce how to use vision foundation models to transfer geometric knowledge for calibrating and restoring the true embedding distributions of tail classes (Section 3.1). Then, we propose a more concise geometric knowledge-based uncertainty representation layer for end-to-end training of long-tail classifiers (Section 3.2). **It is worth noting that** our approach does not involve fine-tuning vision foundation models; it only requires calibration of embedding distributions to train a better long-tail classifier. Therefore, our method can also directly utilize pre-trained CLIP models (such as CLIP-Adapter Gao et al. (2024)) to generate image embeddings of long-tailed distributions.

### 3.1 RECOVERING AND CALIBRATING THE EMBEDDING DISTRIBUTION OF THE TAIL CLASS

Assuming ImageNet-1k is used as an external knowledge base, denoted by $\mathrm{IN}_1, \ldots, \mathrm{IN}_{1000}$ representing 1000 categories. Given a tail class $C_i$ in the long-tail dataset, the $p$-dimensional image embeddings of this class are extracted using a vision foundation model (CLIP/DINOv2) as $Z_i = [Z_i^1, \ldots, Z_i^m] \in \mathbb{R}^{p \times m}$, where $m$ represents the number of samples. The prototype of tail class $C_i$ is computed as the mean of each dimension of the image embeddings:

$$\mu_i = \left( \sum_{j=1}^m Z_i^j \right) / m, \tag{4}$$

and similarly for each category in ImageNet-1k.

Using cosine distance to measure inter-class similarity, let's assume that the class most similar to tail class $C_i$ in ImageNet-1k is $\mathrm{IN}_j$. We extract the image embeddings corresponding to $\mathrm{IN}_j$ using a vision foundation model as $Z_{\mathrm{IN}_j} = [Z_{\mathrm{IN}_j}^1, \ldots, Z_{\mathrm{IN}_j}^n] \in \mathbb{R}^{p \times n}$. The covariance matrix of the embedding distribution for class $\mathrm{IN}_j$ is estimated as:

$$\Sigma_{\mathrm{IN}_j} = E\left[ \frac{1}{n} \sum_{k=1}^n Z_{\mathrm{IN}_j}^k \left( Z_{\mathrm{IN}_j}^k \right)^T \right] = \frac{1}{n} Z_{\mathrm{IN}_j} \left( Z_{\mathrm{IN}_j} \right)^T \in \mathbb{R}^{p \times p}. \tag{5}$$

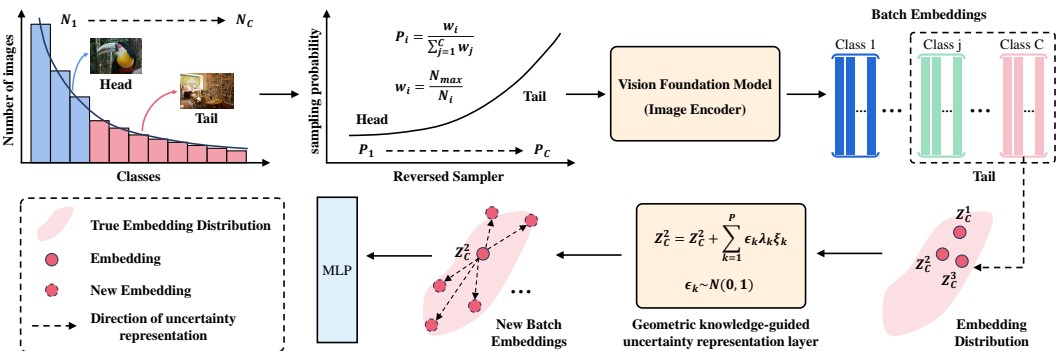

Figure 7: Complete Training Procedure Using GUR. At each iteration, inverse frequency sampling is conducted first, followed by passing the embeddings extracted by the foundation model to GUR, and finally, classification is performed by MLP.

Performing eigenvalue decomposition on the matrix $\Sigma_{\text{IN}_j}$ yields $p$ eigenvalues $\lambda_1 \geq \cdots \geq \lambda_P$ and their corresponding $p$-dimensional eigenvectors $\xi_1, \ldots, \xi_p$. The eigenvectors and eigenvalues respectively provide the direction and magnitude for augmenting and recovering the distribution of the tail class, as supported by the experimental results in Section 2.2.

Specifically, we aim to restore the true distribution of tail classes as much as possible by generating new samples for them in the embedding space. Firstly, we conduct $N$ random linear combinations of the eigenvectors $\xi_1, \ldots, \xi_P$ to obtain $N$ distinct vectors

$$\beta = \sum_{k=1}^{p} \epsilon_k \lambda_k \xi_k \in \mathbb{R}^p, \tag{6}$$

where $\epsilon_k$ follows the standard Gaussian distribution $N(0,1)$. Next, using the existing samples $Z_i^1$ of tail class $C_i$ as the center, we obtain $n$ new samples by $Z_i^1 + \beta$. The same operation is applied to the remaining $m - 1$ samples of tail class $C_i$, resulting in a total of $n \times m$ new samples to restore the true distribution of tail class $C_i$ as much as possible. In this work, we ensure that the number of samples for the augmented tail class is consistent with the number of samples for the most frequent class. For example, in CIFAR-10-LT, the sample number of the augmented tail class is set to 5000.

## 3.2 Geometrically Guided Uncertainty Representation Layer (GUR)

Generating new samples is the most direct method to help the tail class recover its true distribution, but it may not be very practical in engineering applications. This is because training the classifier end-to-end becomes impossible until calibration is performed on the tail class. Therefore, in the following, we propose a geometric knowledge-based uncertainty representation layer, which not only achieves tail class calibration but also ensures that the model can be trained end-to-end.

Before training the long-tailed classifier using GUR, we pre-extracted the geometric shapes (including eigenvectors and eigenvalues) of each category in the external knowledge base using a vision foundation model and computed the embedding prototypes for each category. Similarly, we used the vision foundation model to extract feature embeddings for the tail classes in the long-tail dataset and calculated the embedding prototypes. We then matched each tail class to the most similar category in the knowledge base based on the cosine distance between the prototypes. This entire process is performed before using GUR, allowing us to select the already matched category's eigenvectors/eigenvalues for each tail class during the training of the long-tailed classifier.

Hereafter, we no longer generate additional samples for tail classes but instead learn the classifier directly from the long-tailed data. Therefore, to maintain a balanced optimization, we employ a mechanism of inverse sampling at each iteration Zhou et al. (2020). That is, if a class has more samples, its probability of being sampled is lower. Assuming there are $C$ classes, each with a sample count of $N_i$, the sampling probability for class $i$ can be calculated as

$$P_i = \frac{w_i}{\sum_{j=1}^{C} w_j}, \text{ where } w_i = \frac{N_{\max}}{N_i}, N_{\max} = \max\{N_1, \ldots, N_C\}. \tag{7}$$

Figure 7 clearly illustrates the training process. A balanced mini-batch of training data is obtained through the reverse sampling process. However, in this batch, samples belonging to tail classes may be repeatedly sampled, lacking diversity. Therefore, we characterize each tail class sample in a batch with uncertainty representation to enhance information and calibrate tail classes. Specifically, given a tail class sample $Z_C^i$, after passing through the uncertainty representation layer guided by geometric knowledge (GUR), $Z_C^i$ is represented as a new embedding:

$$Z_C^i = Z_C^i + \sum_{k=1}^{P} \epsilon_k \lambda_k \xi_k, \text{where } \epsilon_k \sim N(0,1). \tag{8}$$

Finally, we employ a simple one-layer MLP to classify the long-tailed data, resulting in a very small number of learnable parameters. Since our method only calibrates the embedding distribution, it can be easily combined with other foundation models for long-tail classification.

## 4 EMPIRICAL STUDY

### 4.1 DATASETS AND EVALUATION METRICS

We evaluate our proposed GUR on four long-tailed benchmark datasets, including CIFAR-10-LT, CIFAR-100-LT Cui et al. (2019), ImageNet-LT Liu et al. (2019), and Places-LT Liu et al. (2019). The imbalance factor (IF) is defined by $\max_k\{n_k\}/\min_k\{n_k\}$, where $n_k$ is the number of samples in the $k$-th class. We conduct experiments on CIFAR-10 &100-LT with IF of 200, 100, 50, and 10. ImageNet-LT is the long-tailed version of ImageNet-2012, with an imbalance factor of 256, consisting of $115.8k$ images across 1000 categories. Places-LT contains $62.5k$ images from 365 classes, from a maximal 4980 to a minimum of 5 images per class. The Top-1 accuracy on the test set is used as the performance metric for the models.

### 4.2 IMPLEMENTATION DETAILS AND COMPARISONS TO EXISTING METHODS

We calibrate the embedding distributions extracted from CLIP Radford et al. (2021), BALLAD Ma et al. (2021), and DINOv2 Oquab et al. (2023) using GUR and train a single-layer MLP for long-tailed classification. We use the SGD optimizer with a learning rate of 0.001 on all datasets. For CIFAR-10 &100-LT, we set the batch size to 64 and trained for 30 epochs. For ImageNet-1k and Places-LT, we set the batch size to 1024 and trained for 10 epochs. It is worth noting that on ImageNet-LT, we employed GUR to transfer knowledge from head classes to tail classes, while on other datasets, we used ImageNet as an external knowledge base.

We particularly focus on comparing **knowledge transfer-based methods** in the long-tailed recognition domain, including OFA Chu et al. (2020), GistNet Liu et al. (2021), CMO Park et al. (2022), FDC Ma et al. (2024a), H2T Li et al. (2024), and FUR Ma et al. (2024b). Additionally, we also compare with **other state-of-the-art methods**, including MiSLAS Zhong et al. (2021), ResLT Cui et al. (2022), and RIDE+CR Ma et al. (2023b), as well as **foundation model fine-tuning methods** CoOp Zhou et al. (2022), CLIP-Adapter Gao et al. (2024), Tip-Adapter-F Zhang et al. (2022), LPT Dong et al. (2022), Decoder and LIFT Shi et al. (2023).

### 4.3 RESULTS ON CIFAR-10-LT AND CIFAR-100-LT

The experimental results are summarized in Table 1. Our method has leaped forward on CIFAR-10-LT and CIFAR-100-LT. Particularly in CIFAR-10-LT, the performance of DINOv2+MLP+GUR surpasses existing state-of-the-art methods at different IF settings: by **17.1%** over FUR at IF 200, **13.6%** over FUR at IF 100, and **11%** over FDC at IF 50. Similarly, in CIFAR-100-LT, at IF of 200, 100, and 50, DINOv2+MLP+GUR outperforms the leading long-tailed recognition method CLIP-Adapter by **21%**, **21.8%**, and **24.2%**, respectively. This significant performance enhancement stems not only from the exceptional performance of the foundation models themselves but also from the outstanding enhancing effect of GUR in long-tailed scenarios, which makes our approach markedly superior to base model fine-tuning methods. For instance, GUR enables CLIP+MLP to achieve performances of **27%**, **29.3%**, and **30.2%** on CIFAR-100-LT at different IF settings.

Table 1: Comparison on CIFAR-10-LT and CIFAR-100-LT. The accuracy (%) of Top-1 is reported. The best and second-best results are shown in **underlined bold** and **bold**, respectively.

| Dataset | Backbone | Pub. | CIFAR-10-LT | | | CIFAR-100-LT | | |
|---|---|---|---|---|---|---|---|---|
| Imbalance Factor (IF) | - | - | 200 | 100 | 50 | 200 | 100 | 50 |
| Cross Entropy | ResNet-32 | - | 65.6 | 70.3 | 74.8 | 34.8 | 38.2 | 43.8 |
| **State-of-the-art long-tail knowledge transfer methods** | | | | | | | | |
| OFA Chu et al. (2020) | ResNet-32 | ECCV 2020 | 75.5 | 82.0 | 84.4 | 41.4 | 48.5 | 52.1 |
| CMO Park et al. (2022) | ResNet-32 | CVPR 2022 | - | - | - | - | 50.0 | 53.0 |
| FDC Ma et al. (2024a) | ResNet-32 | TMM 2024 | 79.7 | 83.4 | 86.5 | 45.8 | 50.6 | 54.1 |
| GCL+H2T Li et al. (2024) | ResNet-32 | AAAI 2024 | 79.4 | 82.4 | 85.4 | 45.2 | 48.9 | 53.8 |
| FUR Ma et al. (2024b) | ResNet-32 | IJCV 2024 | 79.8 | 83.7 | 86.2 | 46.2 | 50.9 | 54.1 |
| **Other state-of-the-art methods** | | | | | | | | |
| MiSLAS Zhong et al. (2021) | ResNet-32 | CVPR 2021 | - | 82.1 | 85.7 | - | 47.0 | 52.3 |
| RIDE (4*) + CR Ma et al. (2023b) | ResNet-32 | CVPR 2023 | - | - | - | - | 49.8 | 59.8 |
| RIDE + H2T Li et al. (2024) | ResNet-32 | AAAI 2024 | - | - | - | 46.6 | 51.4 | 55.5 |
| **Fine-tuning foundation model** | | | | | | | | |
| BALLAD Ma et al. (2021) | ViT-B/16 | - | - | - | - | - | 77.8 | - |
| CLIP (Zero-Shot) Radford et al. (2021) | ViT-B/16 | ICML 2022 | 73.8 | 73.8 | 73.8 | 52.2 | 52.2 | 52.2 |
| CoOp Zhou et al. (2022) | ViT-B/16 | IJCV 2022 | 74.4 | 76.1 | 78.6 | 54.3 | 54.6 | 57.8 |
| CLIP-Adapter Gao et al. (2024) | ViT-B/16 | IJCV 2024 | 72.4 | 75.6 | 79.7 | 58.9 | 61.7 | 62.5 |
| LIFT Shi et al. (2023) | ViT-B/16 | ICML 2024 | - | - | - | - | **80.3** | **82.0** |
| **Calibrating embedding distributions (Ours)** | | | | | | | | |
| CLIP + MLP Radford et al. (2021) | ViT-B/16 | ICML 2022 | 82.4 | 84.7 | 88.5 | 47.5 | 49.6 | 51.4 |
| + GUR | ViT-B/16 | - | **94.4** | **94.6** | **94.8** | **74.5** | 78.9 | 81.6 |
| DINOv2 + MLP Oquab et al. (2023) | ViT-B/16 | TMLR 2024 | 90.3 | 92.1 | 93.4 | 70.7 | 76.2 | 79.2 |
| + GUR | ViT-B/16 | - | **96.9** | **97.3** | **97.5** | **79.9** | **83.5** | **86.7** |

## 4.4 RESULTS ON IMAGENET-LT AND PLACES-LT

Table 2 demonstrates the significant enhancement effect of GUR on CLIP and BALLAD. On the Tail subsets of ImageNet-LT and Places-LT, GUR enables CLIP to achieve performance gains of **24.7%** and **23.2%** respectively. Even though BALLAD is specifically designed for long-tailed scenarios, GUR still improves the overall performance of BALLAD by **2.8%** and **2.4%** on these two datasets. We observe that GUR sometimes reduces the performance of the Head subset, but its ability to significantly improve the performance of the Middle and Tail subsets leads to a smaller bias while enhancing the overall performance of the model. We would like to explain this from the perspective of C2AM Wang et al. (2022). Since the MLP in CLIP+MLP is trained directly on long-tailed data, the resulting decision space is pathological. Specifically, C2AM visualized the weight norms of each class in classifiers trained on long-tail data and observed that the weight norms were highly imbalanced, leading to a pathological decision boundary where the decision space for tail classes is severely compressed. In summary, the good performance of CLIP+MLP on head classes comes at the cost of severely impairing the performance of middle and tail classes. For example, CLIP+MLP achieves 51.4% accuracy on head classes in Places-LT but only 21.3% accuracy on tail classes. The success of GUR is attributed to both the strong capability of the foundation model and several phenomena discovered exclusively on the base model. The collision and fusion of the foundation model with prior knowledge make our method significantly superior to existing methods.

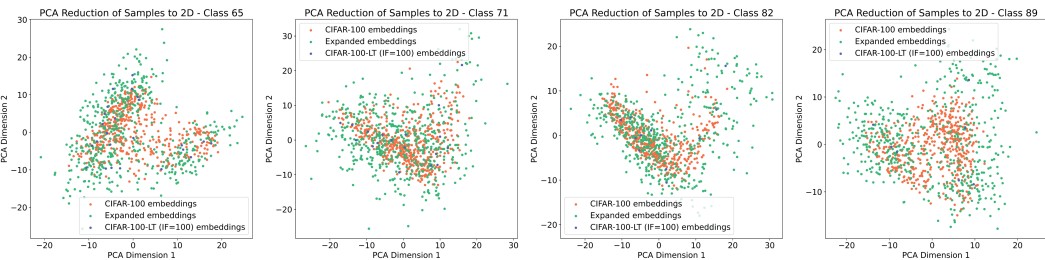

Figure 8: The Calibration Effectiveness of Tail Class Embedding Distributions.

Table 2: Comparisons (Top-1 accuracy (%)) with state-of-the-art methods on ImageNet-LT and Places-LT. The best and the second-best results are shown in **underline bold** and **bold**, respectively.

| Methods | Pub. | ImageNet-LT | | | | Places-LT | | | |
|---|---|---|---|---|---|---|---|---|---|
| | | Head | Middle | Tail | Overall | Head | Middle | Tail | Overall |
| **State-of-the-art long-tail knowledge transfer methods** | | | | | | | | | |
| | | | ResNext-50 | | | | ResNet-152 | | |
| OFA Chu et al. (2020) | ECCV 2020 | 47.3 | 31.6 | 14.7 | 35.2 | 42.8 | 37.5 | 22.7 | 36.4 |
| GistNet Liu et al. (2021) | ICCV 2021 | 52.8 | 39.8 | 21.7 | 42.2 | 42.5 | 40.8 | 32.1 | 39.6 |
| RIDE + CMO* Park et al. (2022) | CVPR 2022 | 66.4 | 53.9 | 35.6 | 56.2 | - | - | - | - |
| RIDE + H2T Li et al. (2024) | AAAI 2024 | 67.6 | 54.9 | 37.1 | 56.9 | 43.0 | 42.6 | 36.3 | 41.4 |
| FUR Ma et al. (2024b) | IJCV 2024 | 65.4 | 52.2 | 37.8 | 55.5 | - | - | - | - |
| **Other state-of-the-art methods** | | | | | | | | | |
| MiSLAS Zhong et al. (2021) | CVPR 2021 | 62.5 | 49.8 | 34.7 | 52.3 | 39.6 | 43.3 | 36.1 | 40.4 |
| DSB + RIDE Ma et al. (2023a) | ICLR 2023 | 68.6 | 54.5 | 38.5 | 58.2 | - | - | - | - |
| ResLT Cui et al. (2022) | TPAMI 2023 | 59.4 | 51.0 | 41.3 | 52.7 | 40.3 | 44.4 | 34.7 | 41.0 |
| RIDE (4*) + CR Ma et al. (2023b) | CVPR 2023 | 68.5 | 54.2 | 38.8 | 57.8 | - | - | - | - |
| **Fine-tuning foundation model** | | | | | | | | | |
| | | | ViT-B/16 | | | | ViT-B/16 | | |
| CLIP (Zero-Shot) Radford et al. (2021) | ICML 2022 | 67.7 | 66.5 | 66.4 | 67.0 | 34.7 | 37.9 | 44.7 | 39.2 |
| CoOp Zhou et al. (2022) | IJCV 2022 | 74.6 | 68.4 | 65.6 | 70.4 | 41.8 | 38.5 | 44.3 | 40.9 |
| Tip-Adapter-F Zhang et al. (2022) | ECCV 2022 | 74.2 | 73.2 | 61.1 | 71.8 | 38.3 | 45.1 | 33.4 | 40.2 |
| LPT Dong et al. (2022) | ICLR 2023 | - | - | - | - | 49.3 | 52.3 | 46.9 | 50.1 |
| Decoder Wang et al. (2024) | IJCV 2024 | - | - | - | 73.2 | - | - | - | 46.8 |
| LIFT Shi et al. (2023) | ICML 2024 | 80.2 | **76.1** | **71.5** | **77.0** | **51.3** | **52.2** | **50.5** | **51.5** |
| **Calibrating embedding distributions (Ours)** | | | | | | | | | |
| CLIP + MLP Radford et al. (2021) | ICML 2022 | **84.5** | 56.8 | 35.7 | 64.6 | **51.4** | 31.6 | 21.3 | 36.5 |
| + GUR | - | **80.6** | 71.9 | 60.4 | 73.5 | 42.9 | 40.6 | 44.5 | 42.1 |
| DINOv2 + MLP Oquab et al. (2023) | TMLR 2024 | 80.2 | 68.4 | 52.6 | 70.3 | 40.6 | 41.0 | 33.4 | 39.3 |
| + GUR | - | 80.3 | 75.2 | 69.1 | 76.5 | 45.2 | 43.8 | 42.5 | 44.3 |
| BALLAD Ma et al. (2021) | - | 79.1 | 74.5 | 69.8 | 75.7 | 49.3 | 50.2 | 48.4 | 49.5 |
| + GUR | - | 80.5 | **77.8** | **74.6** | **78.5** | 51.0 | **52.6** | **51.2** | **51.9** |

### 4.5 MORE ADVANTAGES AND ANALYSIS

**Visualization Examples of Tail Class Calibration.** We utilize our method to generate new samples for the tail classes on CIFAR-100 with an IF of 100 and visualize the results. As shown in Figure 8, the green samples generated from a small number of blue samples cover the real distribution (i.e., orange samples) well. It is particularly noteworthy that the geometric shape of the new distribution is very close to that of the real distribution, which strongly validates the rationality of our motivation.

**Fewer Learnable Parameters (M) and Faster Training Speed.** We compared our method with traditional approaches and fine-tuning methods based on the foundation model. As depicted in Figure 9B, our approach demonstrates superior performance while requiring fewer learnable parameters and converging faster.

**An extreme example.** Randomly select one image from each class of CIFAR-100, totaling 100 images. Extract embeddings for the 100 images using CLIP and DINOv2, then compare the performance of the MLP trained before and after using GUR on the test set. Figure 9A illustrates that GUR still plays a significant role.

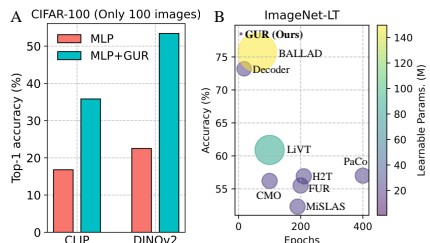

Figure 9: A. Performance of GUR in extreme scenarios. B. Number of Learnable Parameters and Training Speed.

### 5 CONCLUSION

This study discovered three phenomena regarding the transferability of geometric knowledge about embedding distributions, which are only manifested in vision foundation models. Based on these findings, we propose the Geometrically Guided Uncertainty Representation (GUR), achieving calibration of tail class distributions and end-to-end training. GUR demonstrates superior performance while requiring fewer learnable parameters and faster training speed. We believe this work will serve as a typical example of the powerful integration of base models with prior knowledge.

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

# A RELATED WORK

## A.1 CLASS REBALANCING

The extreme imbalance in the number of samples in the long-tailed data prevents the classification model from learning the distribution of the tail classes adequately, which leads to poor performance of the model on the tail classes. Therefore, methods to rebalance the number of samples and the losses incurred per class (i.e., resampling and reweighting) are proposed. Resampling methods are divided into oversampling and undersampling Han et al. (2005); Kang et al. (2019); Wang et al. (2020a); Zhang & Pfister (2021). The idea of oversampling is to randomly sample the tail classes to equalize the number of samples and thus optimize the classification boundaries. The undersampling methods balance the number of samples by randomly removing samples from the head classes. For example, Wang et al. (2020c) finds that training with a balanced subset of a long-tailed dataset is instead better than using the full dataset. In addition, Kang et al. (2019); Zhou et al. (2020) fine-tune the classifier via a resampling strategy in the second phase of decoupled training. Wang et al. (2019) continuously adjusts the distribution of resampled samples and the weights of the two-loss terms during training to make the model perform better. Zang et al. (2021) employs the model classification loss from an additional balanced validation set to adjust the sampling rate of different classes.

The purpose of reweighting loss is intuitive, and it is proposed to balance the losses incurred by all classes, usually by applying a larger penalty to the tail classes on the objective function (or loss function) Huang et al. (2016); Tan et al. (2021); Wang et al. (2017); Ma et al. (2023b;a). Lin et al. (2017) not only assigns weights to the loss of each class but also assigns higher weights to hard samples. Recent studies have shown that the effect of reweighting losses by the inverse of the number of samples is modest Mikolov et al. (2013). Some methods that produce more "smooth" weights for reweighting perform better Ma et al. (2023a), such as taking the square root of the number of samples as the weight. Cui et al. (2019) attributes the better performance of this smoother method to the existence of marginal effects. In addition, Alshammari et al. (2022) proposes to learn the classifier with class-balanced loss by adjusting the weight decay and MaxNorm in the second stage. DSB Ma et al. (2023a) and CR Ma et al. (2023b), for the first time, examined the factors influencing model bias from a geometric perspective and proposed a rebalancing approach.

Although class rebalancing methods are simple to implement, their limitations have been increasingly recognized in recent research Zhang et al. (2023). Class rebalancing methods merely increase the weight of the tail class loss without introducing additional knowledge to assist the tail classes, which often leads to overfitting of the tail classes and significantly compromises the model's generalization performance. Another limitation is that class rebalancing methods often improve tail class performance at the expense of sacrificing head class performance, making it challenging to handle data scarcity issues Zhang et al. (2023); Ma et al.. As a result, more and more research is focusing on information augmentation.

## A.2 STAGE-WISE TRAINING

Decoupling Kang et al. (2019) first proposes to decouple the learning process on long-tail data into feature learning and classifier learning, and it finds that re-learning the balanced classifier can significantly improve the model performance. Further, BBN Zhou et al. (2020) combines the two-stage learning into a two-branch model. The two branches of the model share parameters, with one branch learning using the original data and the other learning using the resampled data. Chu et al. (2020) decomposes the features into class-generic features and class-specific features, and it expands the tail class data by combining class-generic features of the head class with class-specific features of the tail class. Zhong et al. (2021) finds that augmenting data with Mixup in the first stage benefits feature learning and does negligible damage to classifiers trained using decoupling. Zhang et al. (2021) also observes that long-tailed data does not affect feature learning, and it proposes an adaptive calibration function for improving the cross-entropy loss. Jiang et al. (2022) considers the effect of noisy samples on the tail class and adaptively assigns weights to the tail class samples by meta-learning in the second stage. The two-stage training pushes the decision boundary away from the augmented tail class distribution, thereby improving the performance of the tail classes. However, this may lead to excessive bias in the decision boundary and affect the head classes Yin et al. (2019).

## A.3 MODULE IMPROVEMENT

In addition to information enhancement to improve performance from a data perspective, researchers have designed numerous network modules for long-tailed recognition. The methods in this section can be divided into representation learning, classifier design, decoupled training, and ensemble learning. Decoupled training divides the training process into representation learning and classifier learning. LMLE Huang et al. (2016), CRL Dong et al. (2017), KCL Kang et al. (2020) and PaCo Cui et al. (2021) introduce metric learning methods to increase the differentiation of the representation and make the model more robust to data distribution shifts. HFL Ouyang et al. (2016) proposes to hierarchically cluster all classes into leaves of a tree and then improve the generalization performance of the tail classes by sharing the parameters of the parent nodes or similar leaves.

Ensemble learning has shown great potential in long-tailed recognition. BBN Zhou et al. (2020) designed a two-branch network to rebalance the classifier, which is consistent with the idea of decoupled training. To avoid decoupled training damaging the performance of the head class, SimCal Wang et al. (2020a) trained networks with dual branches, one for rebalancing the classifier and the other for maintaining the performance of the head class. ACE Cai et al. (2021), RIDE Wang et al. (2020c) introduced multiple experts with specific complementary capabilities, which led to a significant improvement in the overall performance of the model.

## A.4 HEAD-TO-TAIL KNOWLEDGE TRANSFER

Class rebalancing is inherently unable to handle missing information because no additional information is introduced. Information augmentation aims to improve the performance on tail classes by introducing additional information into the model training. This method is classified into two types: knowledge transfer and data augmentation.

There are four main schemes of knowledge transfer, which are head-to-tail knowledge transfer, model pre-training, knowledge distillation, and self-training. Head-to-tail knowledge transfer aims to transfer knowledge from the head classes to the tail classes to improve the performance of the tail classes. FTL Yin et al. (2019) assumes that the feature distributions of the common and UR classes (i.e., rare classes) have the same variance, so the variance from the head classes is used to guide the feature enhancement of the tail classes. LEAP Liu et al. (2020) transfers the intra-class angle distribution of features to the tail classes and constructs a "feature cloud" centered on each feature to expand the distribution of the tail classes. Similar to the adversarial attack, M2m Kim et al. (2020) proposes to transform some samples from the head class into the tail samples by perturbation-based optimization to achieve tail augmentation. OFA Chu et al. (2020) decomposes the features of each class into class-generic features and class-specific features. During training, the tail class-specific features are fused with the head class-generic features to generate new features to augment the tail classes. GIST Liu et al. (2021) proposes to transfer the geometric information of the feature distribution boundaries of the head classes to the tail classes by increasing the classifier weights of the tail classes. The motivation of the recently proposed CMO Park et al. (2022) is very intuitive, it argues that the images from the head classes have rich backgrounds, so the images from the tail classes can be pasted directly onto the rich background images of the head classes to increase the richness of the tail images. The remaining three types of schemes are relatively few. Cui et al. (2018) first pre-trains the model with all the long-tailed samples, and then fine-tunes the model on a balanced training subset. Yang & Xu (2020) proposes to pre-train the model with self-supervised learning and perform standard training on the long-tailed data. LST Hu et al. (2020) utilizes knowledge distillation to overcome catastrophic forgetting in incremental learning. FDC Ma et al. (2024a) provides detailed experimental evidence for the first time that similar classes have similar distribution statistics, and proposes transferring the variance from head classes to tail classes. FUR Ma et al. (2024b) found that if two classes are very similar, then their distribution shapes are also very similar. Therefore, it proposes transferring the geometric shape of the head class distribution to the tail class to generate new samples for the tail class.

Data augmentation in long-tailed recognition improves the performance of tail classes by improving conventional data augmentation methods. MiSLAS Zhong et al. (2021) suggests adopting mixup to augment feature learning, while not using mixup in classifier learning. FASA Zang et al. (2021) proposes to generate features based on Gaussian prior and evaluate weak classes on a balanced

dataset to adjust the sampling rate. MetaSAug Li et al. (2021) generates augmented features for tail classes with ISDA.

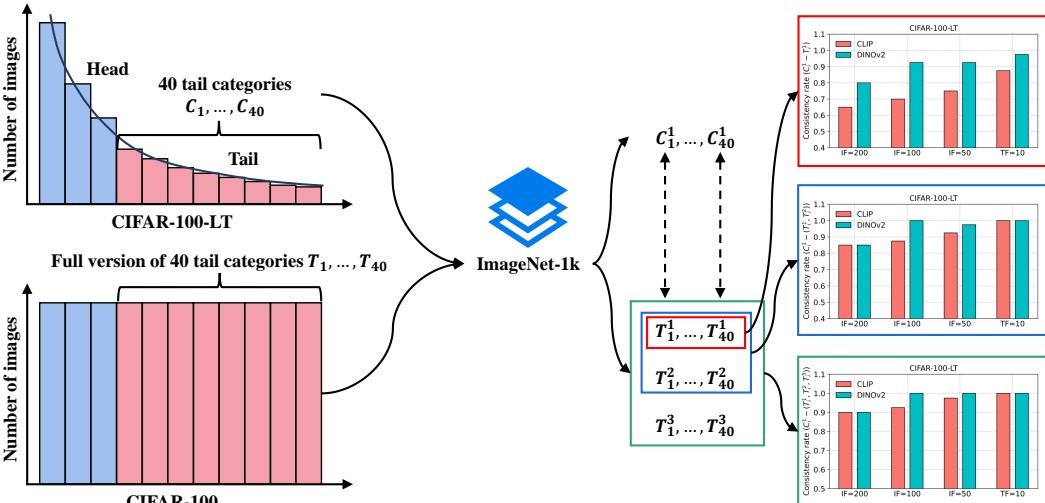

Figure 10: First, match the most similar classes in ImageNet-1k for each of the 40 tail classes of CIFAR-100-LT. Then find the full version of the 40 tail classes in CIFAR-100, and then match the first similar, second similar, and third similar classes in ImageNet-1k for each of the 40 classes in the full version. Check whether the long-tailed version and the full version of a category can be matched to the same class.

## B  MATCHING SIMILAR CLASSES FOR TAIL CLASS DISTRIBUTION RESTORATION

We demonstrate in Figure 10 how to verify whether a long-tailed version and a complete version of a class can be matched to the same category in ImageNet-1k. The similarity between classes is measured by the cosine distance between prototypes of class image embeddings, all of which are extracted by CLIP and DINOv2. We examine whether $C_1^1$ matches $T_1^1$, $C_2^1$ matches $T_2^1$, and so forth up to $C_{40}^1$ matching $T_{40}^1$, and tally the proportion of matches out of 40, resulting in the first bar chart in Figure 10. Let $m = 0$, and check whether $C_1^1$ is included in $T_1^1$ and $T_2^1$. If it is, increment m by 1. Continue checking whether $C_{40}^1$ is included in $T_{40}^1$ and $T_{40}^1$, and calculate $m/40$, resulting in the second bar chart. The third bar chart follows a similar procedure as the second.

