# OpenReview forum: "Vision Foundation Models Bridge the Geometric Knowledge Across Domains for Long-Tailed Recognition"
_ICLR.cc/2025/Conference — Submitted to ICLR 2025_

### Official Review · Reviewer_BWCu · 2024-10-24

**Soundness:** 2
**Presentation:** 3
**Contribution:** 2
**Rating:** 5
**Confidence:** 4

**Summary:**

This paper introduces the Geometrically Guided Uncertainty Representation (GUR), a method that enhances long-tailed recognition by transferring geometric knowledge from head classes to tail classes using foundation models like CLIP and DINOv2. GUR improves tail class embeddings without generating new samples, achieving notable performance gains across various benchmarks.

**Strengths:**

- The overall writing of the paper is smooth and easy to understand.
- GUR allows for end-to-end training, making it more practical for real-world deployment compared to methods requiring additional data generation.

**Weaknesses:**

- The selection of the external knowledge base is crucial for the proposed method, making it a significant limitation.
- The method struggles with complex datasets and has not been tested on the widely-used long-tailed dataset iNaturalist 2018. The reviewer anticipates that the method would perform poorly on iNaturalist, which has over 8,000 categories—far more than ImageNet-1k's 1,000 classes. Additionally, iNaturalist is a fine-grained dataset, leading to many categories being mapped to the same class, thereby confusing head and tail classes and exacerbating the long-tail problem.
- The direct use of ImageNet-1k as the external knowledge base suits the three smaller datasets used in this study, particularly since the classes of CIFAR-10-LT and CIFAR-100-LT are subsets of ImageNet-1k, meaning its rich features sufficiently cover the target datasets. The reviewer believes that the positive results on these datasets are more related to the comprehensive features of ImageNet-1k itself rather than the proposed method. Moreover, the paper does not provide results of fine-tuning the networks pre-trained on ImageNet-1k on the corresponding datasets.
- For ImageNet-LT, the authors use head class knowledge as the external knowledge base to enhance the tail classes, seemingly to avoid using the full ImageNet-1k dataset. If this is the approach, why not use head class data to enhance tail classes across all datasets?

**Questions:**

Using head class knowledge as the external knowledge base on ImageNet-LT could make similar head and tail classes more similar in the feature space, potentially intensifying the long-tail effect by increasing the likelihood of tail classes being misclassified as similar head classes. How does the method address this issue?

---

> ### Author Response · Authors · 2024-11-27
> **To Reviewer BWCu**
>
> Thank you for your professional feedback. Your concerns are critical, and we considered them during the development of this work.
>
> First, we believe that leveraging external knowledge bases is a promising direction, especially in an era of rapid advancements in large models. Enhancing specific scenarios with external knowledge bases is both meaningful and practical.
>
> Second, fine-tuning models pretrained on ImageNet often performs poorly on long-tailed datasets, significantly underperforming the results presented in this paper. This observation has already been reported in existing research.
>
> Lastly, using head classes to assist tail classes without relying on external datasets is feasible and effective, as we have verified. The paper *Ma, Yanbiao, et al. "Geometric Prior Guided Feature Representation Learning for Long-Tailed Classification." International Journal of Computer Vision (2024): 1-18* explores the feasibility of leveraging head classes to enhance tail classes.
>
> We sincerely appreciate the time and effort you have dedicated to reviewing our work. **Wishing you a wonderful day.**

---

### Official Review · Reviewer_9vH3 · 2024-11-01

**Soundness:** 3
**Presentation:** 2
**Contribution:** 2
**Rating:** 5
**Confidence:** 3

**Summary:**

This work is motivated by the empirical observation that vision foundation models can capture inter-class similarity. Based on this insight, the authors build a knowledge base using a balanced image dataset and apply a geometric distribution to assist rare classes in long-tailed recognition (LTR). They introduce a Geometrically Guided Uncertainty Representation (GUR) layer for end-to-end training. The experiments demonstrate performance improvements across multiple long-tailed benchmarks compared to prior state-of-the-art (SOTA) methods.

The concept of using vision foundation models to shift imbalanced distribution is intriguing and valuable. However, the paper's organization needs refinement, and some aspects of the experimental setup, particularly the comparisons with methods trained from scratch, may not be entirely fair.

**Strengths:**

- The motivation for the work is clear, technically sound, and supported by toy experiments that illustrate the authors' observations.
- The proposed method is straightforward and appears effective, with an easy-to-follow approach.

**Weaknesses:**

- Potential Data Leakage (L137): The use of the full dataset rather than specifically long-tailed data could introduce label leakage, especially since CIFAR-LT is sampled from CIFAR. How do the authors accurately obtain the true distribution of tail classes in naturally imbalanced datasets, such as iNaturalist, where such knowledge might not be available?

- Section 2 Organization: Section 2 lacks clarity, as it includes a number of prerequisite trials that could be streamlined. I suggest focusing on key observations in the main paper and moving intermediate findings to the supplementary material to improve readability.

- Fairness of Experimental Comparisons: The comparisons with methods that train from scratch seem unfair due to the strong prior used from the VLM. It might be more appropriate to compare with methods that also incorporate strong priors or clarify the differences in setup.

**Questions:**

- Balanced Prior and OOD Concerns (Sec. 3): The balanced prior is based on a specific dataset (e.g., ImageNet-1k in L311). How does the method handle out-of-distribution cases in the knowledge base? For instance, if a long-tailed dataset is cat-focused, but the knowledge base consists of balanced dog data, what challenges might arise? Do the authors consider and address these potential OOD issues?

- Data Augmentation Clarification (L359): The authors argue that generating new samples is impractical, yet data augmentation is common in LTR to supplement the dataset. Although GUR adjusts the latent space, does this mean that no pseudo-samples (e.g., mixup) are generated during training? A comparison of these strategies would be helpful.

- Performance Variation: The method shows substantial improvements on CIFAR-LT but limited gains on ImageNet-LT and Places-LT. Can the authors elaborate on these results? Is this discrepancy due to potential OOD mapping, or are there other factors involved?

- Calibration Evidence: Since the authors discuss calibration improvements, quantitative evidence supporting the calibration benefits of the proposed method would strengthen the findings.

**Details Of Ethics Concerns:**

- Balanced Prior and OOD Concerns (Sec. 3): The balanced prior is based on a specific dataset (e.g., ImageNet-1k in L311). How does the method handle out-of-distribution cases in the knowledge base? For instance, if a long-tailed dataset is cat-focused, but the knowledge base consists of balanced dog data, what challenges might arise? Do the authors consider and address these potential OOD issues?

- Data Augmentation Clarification (L359): The authors argue that generating new samples is impractical, yet data augmentation is common in LTR to supplement the dataset. Although GUR adjusts the latent space, does this mean that no pseudo-samples (e.g., mixup) are generated during training? A comparison of these strategies would be helpful.

- Performance Variation: The method shows substantial improvements on CIFAR-LT but limited gains on ImageNet-LT and Places-LT. Can the authors elaborate on these results? Is this discrepancy due to potential OOD mapping, or are there other factors involved?

- Calibration Evidence: Since the authors discuss calibration improvements, quantitative evidence supporting the calibration benefits of the proposed method would strengthen the findings.

---

> ### Author Response · Authors · 2024-11-27
> **To Reviewer 9vH3**
>
> Thank you very much for your professional feedback. Your concerns are extremely important. However, I have been very busy recently and will respond to your questions and concerns at a later time, even if this paper is not accepted. Once again, I sincerely appreciate the time and effort you have dedicated. **Wishing you a wonderful day ahead.**

---

### Official Review · Reviewer_1eGV · 2024-11-03

**Soundness:** 1
**Presentation:** 1
**Contribution:** 2
**Rating:** 3
**Confidence:** 4

**Summary:**

This paper studies to restore the true distribution of rare classes using the shape and size of the most similar common class. The motivation is that similar classes have similar shape of the latent-feature distribution.  Instead of synthesizing new image examples, this paper follows the idea of augmenting the latent features of rare classes with the geometric information from the nearest-neighbor common class.  The shape information is characterized by the bases (eigen vectors) of the feature space spanned within a category.  The size information is captured by the summation of eigen values.  Features are augmented by adding the linear combinations of these bases from the matched common category.  This paper shows significant performance gain in the tail classes on CIFAR10-LT and CIFAR100-LT datasets.

**Strengths:**

1. The idea of augmenting latent features of rare classes to tackle long-tailed distribution is interesting.

2. The performance gain in the tail classes on CIFAR10-LT and CIFAR100-LT is significant.

**Weaknesses:**

1. My main concern is a lack of theoretical justification for the proposed method.  The authors claim that latent-feature distribution of similar classes have similar shape and size, however, the proposition is not theoretically proved.  The only evidence provided by the authors is the similarity plots shown in Figure 3, 4 and 5.  As mentioned in Section 2.2.1 that small models do not associate geometric shapes across datasets, I am concerned if the results shown in Figure 3, 4, and 5 only apply to the selected datasets (CIFAR, Caltech-101 and ImageNet) and models (DINOv2 and CLIP).

2. The performance of head classes drop significantly on ImageNet-LT and Places-LT (Table 2)

3. The definition of size (Eqn 2) is weird.  Let say we have two pairs of $\lambda_1, \lambda_2$: $ 0.5, 0.5$ and $0.9, 0.1$.  The former should have larger area then the later.  But, they instead have the same size according to Eqn 2.

4. Figure 3 doesn't support the correlation between shape similarity and categorical similarity.  Nearly 20% of the rows have high similarity of shapes to the top-48 nearest-neighbor classes.

5. Figure 5 doesn't support the correlation between size similarity and categorical similarity.  For each query class, the author should plot the size ratio (y-axis) w.r.t the feature similarity (x-axis).  A query class may have a size ratio of 1 w.r.t the least-similar common class.

6. Figure 6 is confusion.  The definition of "IF" is missing.

**Questions:**

1. Is the proposition valid for other domains, such as aerial/satellite imagery, medical imagery (e.g. MRI), or even object detection dataset (e.g. LVIS)?

2. Why does the performance of the head classes drop in Table 2?

3. Do you have any justification of the definition of size (Eqn 2)?

4. Can you show the same plot as Figure 3 on other-domain dataset (e.g. aerial satellite imagery, or LVIS)?

5. Can you show the same plot as Figure 5 on other-domain dataset (e.g. aerial satellite imagery, or LVIS)?

6. What is "IF" in Figure 6?

---

> ### Author Response · Authors · 2024-11-27
> **To Reviewer 1eGV**
>
> Thank you for your insightful comments. Our work focuses on natural image datasets and is conducted on comprehensive long-tailed benchmark datasets. Since CLIP is trained on natural images, I believe it is unnecessary to explore the phenomenon proposed in this paper on remote sensing images.
>
> Moreover, as a phenomenon, such work is currently difficult to prove theoretically because deep neural networks are inherently opaque. Similar to discovering phenomena and patterns from numerous natural results, theoretical proof requires advancements in fundamental science. Only if breakthroughs are made in the foundational theories of deep neural networks will it be possible to provide theoretical proof for certain phenomena.
>
> In Figure 6, "IF" represents the degree of imbalance in the dataset, and its full name is "Imbalance Factor."
>
> In conclusion, we sincerely appreciate the time and effort you have dedicated to reviewing our work. **Wishing you a wonderful day.**

---

### Official Review · Reviewer_ZSms · 2024-11-05

**Soundness:** 3
**Presentation:** 3
**Contribution:** 3
**Rating:** 8
**Confidence:** 5

**Summary:**

This paper tackles the problem of long-tailed recognition problem in computer vision. This study identifies that vision foundation models can correlate inter-class similarity with the geometric shapes of class distribution in cross-domain scenarios. Specifically, when two classes are highly similar, their embedding distributions also exhibit geometric characteristics. The research introduces the Geometrically Guided Uncertainty Representation (GUR) Layer, designed for long-tailed recognition tasks to calibrate and enhance the embedding distribution of tail classes, ultimately facilitating the learning of an unbiased MLP classifier. Experiments on various long-tailed benchmark datasets demonstrate that GUR significantly improves the performance of long-tailed recognition.

**Strengths:**

1. The proposed method does not require fine-tuning of the foundation model, which significantly enhances its efficiency and ease of training. By avoiding the need for fine-tuning, this approach is more computationally efficient and accessible, especially in scenarios where resources are limited.

2. The description of the method is clear and easy to understand.

3. The experimental comparisons are thorough, including a wide range of relevant methods for evaluation. The paper compares the proposed method against various related approaches, both those that utilize foundation models and those that do not.

**Weaknesses:**

1. The paper utilizes embedding geometry from an external dataset (ImageNet) to support the claim that the foundation model can bridge geometric structures across domains. While I understand the rationale behind this choice. I am curious about the method's performance if the geometry were derived from the head classes of the same dataset, as in the experiment on ImageNet-LT. This approach would avoid the need for external data, lead to a fair comparison with others.

2. The novelty of the proposed method feels somewhat limited, as the core concept closely resembles that of FUR [1]. The primary difference here is the transfer of geometry from an external dataset and the use of foundation models to facilitate this.

[1] Ma, Yanbiao, et al. "Geometric Prior Guided Feature Representation Learning for Long-Tailed Classification." International Journal of Computer Vision (2024): 1-18.

**Questions:**

See strengths and weaknesses.

---

> ### Author Response · Authors · 2024-11-27
> **To Reviewer ZSms**
>
> Thank you for your professional feedback and recognition. Due to my current busy schedule, I have decided not to respond to the reviewers' comments at this time. However, all authors will carefully consider the reviewers' suggestions to further improve this work.
>
> Wishing you a wonderful day!

---

### Meta-Review · Area_Chair_X3y2 · 2024-12-21

**Metareview:**

This paper propose to transfer geometric knowledge from head classes (frequent classes) to tail classes (rare classes), based on the key observation and assumption that vision foundation models can associate inter-class similarity with the similarity of geometric shapes of class distributions in cross-domain scenarios.

This paper received 4 divergent reviews, with ratings 3,8,5,5.  The most negative review questions the theoretical justification about the latent feature shape transfer and also the lack of support or validation from experimental results.  Other negative reviews primarily find the experimental validation unconvincing on more complex datasets beyond CIFAR10 and CIFAR100.  The most positive review likes the efficiency, clarity, and thorough comparisons, but has concerns about limited technical novelty; however, the reviewer also acknowledged OK with the rejection decision post-rebuttal as concerns raised by other reviewers could not be addressed effectively by the authors in time.  Indeed the rebuttal has only brief claims and they are not enough to overturn the negative consensus.

The paper is rejected due to the lack of theoretical justification, limited novelty, questionable experimental effectiveness and generalizability to more complex and naturally long-tailed datasets.

**Additional Comments On Reviewer Discussion:**

The rebuttal is not detailed and timely enough to engage reviewers in discussions.

---

### Decision · Program_Chairs · 2025-01-22

Reject